# Bioinspired graphene membrane with temperature tunable channels for water gating and molecular separation

Jingchong Liu[1], Nü Wang[1], Li-Juan Yu[2], Amir Karton[2], Wen Li[3,4], Weixia Zhang[4], Fengyun Guo[1,4], Lanlan Hou[1], Qunfeng Cheng [1], Lei Jiang[1], David A. Weitz[4,5] & Yong Zhao[1,4]

Smart regulation of substance permeability through porous membranes is highly desirable for membrane applications. Inspired by the stomatal closure feature of plant leaves at relatively high temperature, here we report a nano-gating membrane with a negative temperature-response coefficient that is capable of tunable water gating and precise small molecule separation. The membrane is composed of poly(N-isopropylacrylamide) covalently bound to graphene oxide via free-radical polymerization. By virtue of the temperature tunable lamellar spaces of the graphene oxide nanosheets, the water permeance of the membrane could be reversibly regulated with a high gating ratio. Moreover, the space tunability endows the membrane with the capability of gradually separating multiple molecules of different sizes. This nano-gating membrane expands the scope of temperature-responsive membranes and has great potential applications in smart gating systems and molecular separation.

[1] Laboratory of Bioinspired Smart Interfacial Science and Technology of the Ministry of Education, Beijing Key Laboratory of Bioinspired Energy Materials and Devices, School of Chemistry, Beihang University, Beijing 100901, China. [2] School of Molecular Sciences, The University of Western Australia, 35 Stirling Highway Crawley, Perth, WA 6009, Australia. [3] Laboratory of Polymer Chemistry, Department of Polymer Materials, College of Materials Science and Engineering, Shanghai University, Nanchen Street 333, Shanghai 200444, China. [4] John A. Paulson School of Engineering and Applied Sciences, Harvard University, Cambridge, MA 02138, USA. [5] Department of Physics, Harvard University, Cambridge, MA 02138, USA. Jingchong Liu and Nü Wang contributed equally to this work. Correspondence and requests for materials should be addressed to N.W. (email: wangn@buaa.edu.cn) or to Y.Z. (email: zhaoyong@buaa.edu.cn)

For a number of drought-enduring vegetations in nature (such as cacti), the stomata on their leaves close at relatively high temperatures, effectively reducing the water loss in order to sustain life[1,2]. This biological phenomenon displays a negative temperature-response characteristic from a membrane perspective, that is, the membrane's permeability decreases as the surrounding temperature increases[3–5]. Although this negative temperature-responsive phenomenon is prevalent in nature, most man-made temperature-responsive membranes have the opposite responsive trend where water permeability increases with increasing temperature, i.e., they show a positive temperature-response characteristic[6–9]. This is because most temperature-responsive polymeric materials adopted in gating membranes undergo a coil-globule shrinking transition of their chains at a lower critical solution temperature (LCST)[10–12], and as a result, the membrane pores switch from a closed state below the LCST to an open state above the LCST.

Graphene oxide (GO) is a two-dimensional material consisting of a carbon lattice and oxygen-containing groups[13]. Recently, GO membranes (GOMs) have exhibited superior gating and separation performances due to their unique unimpeded two-dimensional nanochannels and nacre like lamellar structure[14,15]. The ordered brick and mortar assembly of inorganic and/or organic layers in GO-based membranes endows them with strong interfacial interactions and excellent chemical stability[16]. Therefore, GO-based membranes have been successfully applied to gas, molecular, and organic solvent separations[17–30]. Moreover, controlling molecule penetration through GO-based membranes has received considerable attention due to the crucial importance of gatekeepers control the movement of substances, mimicking living organisms[31,32]. Previous studies have demonstrated that water permeance of different kinds of GO-based membranes can be regulated by external fields such as pH, solvent, ion concentration, electric and magnetic fields[20,33–35]. However, these systems rely on specific chemical environments like ionic strength, acid-base reactions and so on, which significantly limit

their scope of practical applications. Temperature controlled gating membrane systems, which are convenient and less dependent on the chemical environment, have more extensive adaptability[36]. However, negative temperature-response graphene-based membranes that can achieve water gating and molecular separation have never been reported.

Here, we construct a negative temperature-response nanogating membrane by covalently grafting poly(N-isopropylacrylamide) (PNIPAM) chains on GO sheets. The water permeance of this membrane varies from $12.4 \, l \, m^{-2} \, h^{-1} \, bar^{-1}$ at 25 °C to $1.8 \, l \, m^{-2} \, h^{-1} \, bar^{-1}$ at 50 °C with a high gating ratio of ~7. Moreover, the tunable lamellar spacing of the GOM enables it to separate small molecules with different sizes by regulating the temperature. This smart membrane shows promise in many fields, such as fluid transport systems, microfluidic chip systems and molecular separation devices.

## Results

**Fabrication of the PNIPAM covalently grafted GOMs.** The temperature-responsive PNIPAM covalently grafted GO membranes(P-GOMs) are based on regulating the layer spacing through tuning the molecular configuration of PNIPAM. The two-step construction process and water gating property of P-GOMs are illustrated in Fig. 1, involving polymerization of N-isopropylacrylamide (NIPAM) on GO sheets to form PNIPAM-grafted GO (P-GO) and assembling P-GO into P-GOMs through pressure-driven filtration. When the temperature is below the LCST of P-GO, the PNIPAM chains on P-GO exhibit a swollen coil structure because of hydrogen bond interaction between amide group of PNIPAM and water molecules. Under this condition, the space between two adjacent GO sheets provides appropriate nanochannels for water transport. If the temperature increases above its LCST, the PNIPAM chain shrinks because the hydrogen bond interaction between the amide group and water molecule is replaced by the intramolecular/

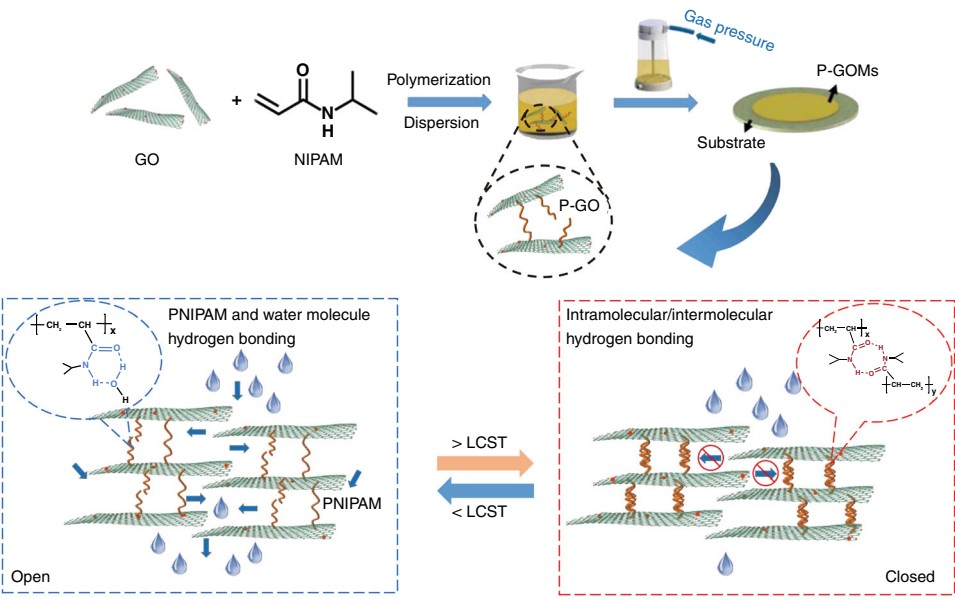

**Fig. 1** Fabrication process of the temperature-responsive membrane and its water gating property. N-isopropylacrylamide (NIPAM) monomers polymerize on graphene oxide (GO) sheets to form PNIPAM-grafted GO (P-GO). Then P-GO assembles into PNIPAM covalently grafted GO membranes (P-GOMs) through pressure-driven filtration of their aqueous dispersion. The water permeance of P-GOMs could be regulated via the environmental temperature (T). P-GOMs have a large water permeance when T is below its lower critical solution temperature (LCST) because of the expanded water flow channel caused by the swollen PNIPAM chains. The water permeance decreases when T is above the LCST because of the narrowed channel of P-GOMs, which is caused by the shrink of PNIPAM chains

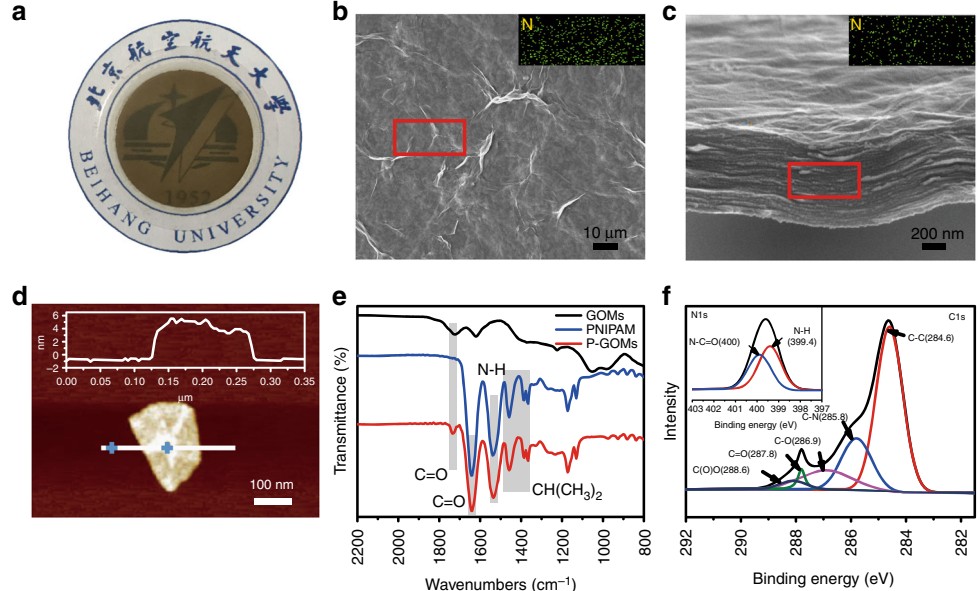

**Fig. 2** Characterization of P-GOMs. **a** Digital photo of P-GOMs. **b, c** Top-view and cross-sectional SEM images of P-GOMs showing a typical wrinkle surface and stacking microstructure. The insets are EDS maps indicating that nitrogen is evenly distributed on both surface and cross-section of P-GOMs. **d** AFM image of a single P-GO sheet with thickness ~5 nm. **e** ATR-FTIR spectra of GOMs, PNIPAM and P-GOMs, illustrating covalent bound between PNIPAM and GO. **f** XPS spectrum of P-GO, of which C1s region fitting into five peaks at 284.6, 285.8, 286.9, 287.8, and 288.6 eV, representing C–C, C–N, C–O, C=O and C(O)O. Inset is N1s region, fitting into two peaks at 399.4 and 400.0 eV, representing amine and amide, respectively, because of the graft of PNIPAM

intermolecular hydrogen bonding of PNIPAM[37]. In this process, the GO sheets are pulled closer because they are covalently tethered to the ends of PNIPAM chains, resulting in a smaller lamellar distance of P-GOMs. Then, the water permeance decreases as water is mostly blocked by P-GOMs. It is thereby feasible to construct P-GOMs with negative coefficient in hydraulic permeability.

The temperature-responsive P-GOMs were obtained by pressure-assisted self-assembly technology[38] with the freeze-dried P-GO (Supplementary Figs. 1a and b). The P-GOMs displayed typical brown color with good optical transparency and uniform surfaces without defects (Fig. 2a and Supplementary Fig. 1c), which was similar to GOMs (Supplementary Fig. 1d)[19]. The surface and cross-section scanning electron microscope images of the P-GOMs (Fig. 2b, c) demonstrate that the P-GOMs have wrinkled surface and stacking microstructure like normal GOMs (Supplementary Figs. 2a and b). The energy dispersive spectroscopy results show that nitrogen element is evenly distributed in both surface and cross-section of P-GOMs (Fig. 2b, c), whereas almost no nitrogen is detected in pristine GOMs (Supplementary Fig. 2). These are side evidences of the existence of PNIPAM between the GO laminates. After the polymerization of NIPAM on GO sheets, the thickness of a single P-GO sheet increased to ~5 nm as shown by the atomic force microscope (AFM) image in Fig. 2d. Hence the thickness of the PNIPAM layer was ~4 nm considering the 0.8 nm thickness of pristine GO sheet (Supplementary Fig. 3). To further examine the interactions between GO and PNIPAM, attenuated total reflectance Fourier transform infrared (ATR-FTIR) measurements of GOMs, pure PNIPAM, and P-GOMs were conducted (Fig. 2e). Besides the stretching peak of oxygen functional groups of GO at 1743 cm$^{-1}$, several new peaks at 1642, 1540, and 1367~1460 cm$^{-1}$ appeared after grafting with PNIPAM[39]. These new peaks are attributed to the C=O stretching (amide I band), the deformation of N-H bond and CH(CH$_3$)$_2$ groups of PNIPAM, respectively[40,41]. These characteristic peaks coincided with the spectrum of pure

PNIPAM. During the reaction, NIPAM not only polymerized on GO surfaces but also formed free PNIPAM through self-polymerization[42]. To eliminate the possibility that the new peaks detected in the ATR-FTIR spectrum of P-GOMs were from the free PNIPAM attached to GO, the filtrate of the P-GOMs construction process was also analyzed by ATR-FTIR (Supplementary Fig. 4). There was no absorption peak of PNIPAM in the spectrum of the filtrate, confirming the strong covalent interaction between PNIPAM and GO. The X-ray photoelectron spectroscopy (XPS) spectra of GO and P-GO clearly indicate the introduction of nitrogen-containing functional groups after modifying GO with PNIPAM (Fig. 2f and Supplementary Fig. 5). The C1s XPS spectrum of P-GO shows curve fittings with a new peak of C–N at the binding energy of 285.8 eV compared with that of GO[43]. More characterization including Raman spectra, and $^1$H nuclear magnetic resonance also demonstrate that PNIPAM chains have been successfully covalently grafted to GO sheets (Supplementary Figs. 6 and 7).

**Temperature-responsive property of the P-GOMs.** As is well known, PNIPAM will undergo a phase transition behavior in water and form aggregates (which makes the solution turbid) when the temperature is above its LCST. This reversible temperature-response property of PNIPAM, as expected, was well retained after it was covalently bound to GO. Figure 3a shows that the P-GO reversibly dispersed and aggregated upon cooling and heating of the aqueous dispersion between 25 °C and 50 °C. This phenomenon was not observed for the GO aqueous dispersion (Supplementary Fig. 8), indicating that GO does not have any temperature-response property. As reflected by the variable-temperature ultraviolet visible (UV-vis) spectrum, the LCST of P-GO was ~30 °C (determined by the temperature point where the absorption value dramatically increased), which was slightly lower than that of pure PNIPAM (~ 32 °C)[44] (Supplementary Fig. 9). This small variation was because GO was partially reduced

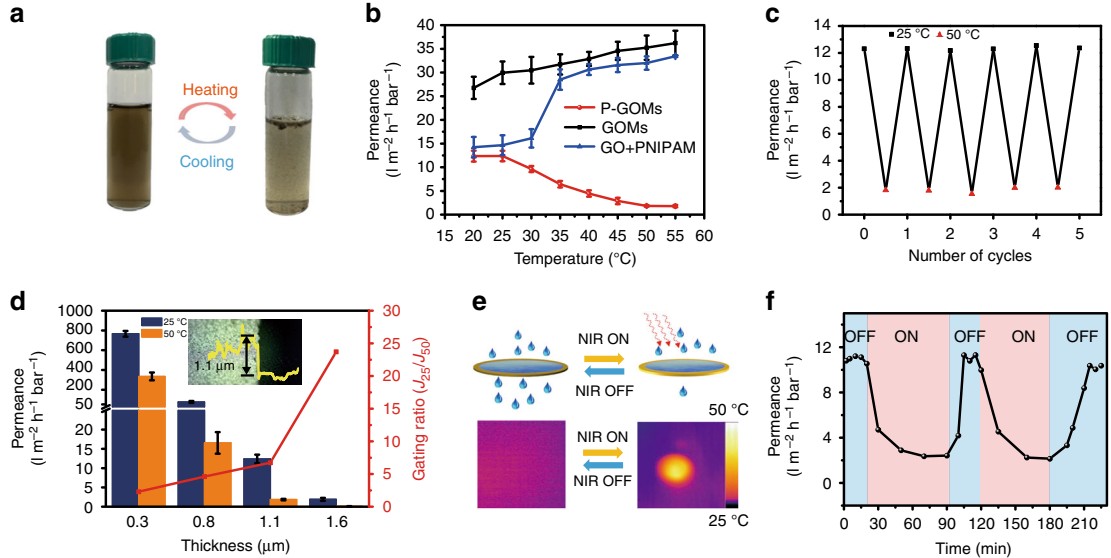

**Fig. 3** Thermal- and photo-gating performances of P-GOMs. **a** P-GO sheets re-dispersed and aggregated upon cooling and heating of the aqueous dispersion between 25 °C and 50 °C, indicating the temperature-responsive property of P-GO. **b** Temperature-dependent water permeance curves of GOMs, P-GOMs and physically blending PNIPAM/GO membranes, P-GOMs show the unique negative temperature-responsive coefficient, which is opposite to the positive coefficient of GOMs and GO/PNIPAM blending membrane. Error bars, s.d. ($n = 3$). **c** Reverse stability test of the temperature-responsive water gating property of P-GOMs. **d** Thickness-dependent water permeance and gating ratio of P-GOMs, 1.1 μm thickness is adopted after balancing the water permeance and high gating ratio. Error bars, s.d. ($n = 3$). **e** Schematic showing the water permeance of P-GOMs decreased after irradiation of near-infrared (NIR). The bottom pictures are thermographic images of P-GOMs before and after irradiation of NIR. **f** The water permeance of P-GOMs decreases when switching on NIR light and returns to initial values when the NIR light is off

after functionalization and thus became a little hydrophobic. As a result, the LCST of PNIPAM decreased when bound to hydrophobic groups[45]. This hypothesis was supported by the results of the XRD, water contact angle, XPS, and Raman characterizations (Supplementary Fig. 10).

These results indicate that PNIPAM is successfully grafted onto GO surfaces by covalent bonds, making it feasible to act as water gating by virtue of the temperature tunable lamellar spaces of the P-GOMs. Then we performed pressure-driven hydraulic permeability experiments to study the temperature-response behavior of P-GOMs. As controls, GOMs and physically blended GO/PNIPAM membrane were also compared. As illustrated in Fig. 3b, the P-GOMs exhibited the negative temperature-response characteristic, which was distinctly different with the GOMs and physically blended GO/PNIPAM membrane. The water permeance of GOMs slightly increased as the temperature went up, owing to the decreased water viscosity[46]. For the physically blended GO/PNIPAM membranes, the permeance had a sharp transition when the temperature changed from 30 °C to 35 °C. Given that the LCST of PNIPAM is ~32 °C, the PNIPAM chains non-covalently attached to GO in the membranes with shrunken conformation expanded the size of the channels at 35 °C compared with that at 30 °C, thus resulting in a larger permeance increase (Supplementary Fig. 11)[36]. For our P-GOMs, however, the permeability decreased with the temperature increasing when the temperature was above 30 °C, showing a negative temperature-response characteristic. The P-GOMs (1.1 μm thickness) had an average water permeance of 12.4 l m$^{-2}$ h$^{-1}$ bar$^{-1}$ at 25 °C, and decreased to 1.8 l m$^{-2}$ h$^{-1}$ bar$^{-1}$ when the temperature increased to 50 °C. The gating ratio of P-GOMs ($J_{25}/J_{50}$, where $J_T$ is the water permeance at temperature $T$ °C) was ~7 and sustained after several cycles (Fig. 3c). Therefore, the P-GOMs can serve as a hydraulic permeability gating membrane with a relatively large negative gating ratio. Furthermore, we studied the thickness effect of P-GOMs on the gating ratio as shown in Fig. 3d. Assuming that water inside the

nanochannels behaves as a viscous flow[29,30,47,48], the water permeated through the P-GOMs can be described by the Hagen–Poiseuille equation:

$$J = \varepsilon \pi r^2 \Delta p / 8 \eta d\tau \tag{1}$$

where $J$ is the water permeance, $\varepsilon$ is the surface porosity, $r$ is the pore radius, $\Delta p$ is the applied pressure, $\eta$ is the water viscosity, $d$ is the membrane thickness and $\tau$ is the tortuosity. According to Equation (1), the water permeance is inversely proportional to the membrane thickness. The thickness variations of P-GOMs with different loading amounts of P-GO aqueous dispersion can be acquired from Supplementary Fig. 12. The water permeance decreased at both of 25 °C and 50 °C as the thickness of P-GOMs increased from 0.3 to 1.6 μm, whereas the gating ratio increased gradually. Although the gating ratio of 23.7 was able to reach when the thickness of P-GOMs was 1.6 μm, the low permeance of only 1.9 l m$^{-2}$ h$^{-1}$ bar$^{-1}$ at 25 °C would restrict its scope of applicability. Thus, the P-GOMs with thickness of 1.1 μm were chosen for the temperature-response gating experiments.

Besides directed thermal tunable permeation, we realized remote control of the membrane permeability via light because of the superior light-to-heat conversion capability of GO. This would be implemented more conveniently in deserved applications[14,32,49]. When the P-GOMs were exposed under 808 nm near-infrared (NIR) light at power density of 0.3 W cm$^{-2}$ in air at an initial temperature of ~25 °C, its temperature rose across the LCST (30 °C) (Fig. 3f). Water has a low absorption at 808 nm NIR while the P-GOMs have a strong absorption, thus the water permeance of the P-GOMs can be regulated by the NIR light (Fig. 3e)[50,51]. As shown in Fig. 3f, the water permeance was ~11.2 l m$^{-2}$ h$^{-1}$ bar$^{-1}$ at ambient temperature (~ 25 °C) without the NIR light. After irradiation, the water permeance of P-GOMs decreased to ~2.8 l m$^{-2}$ h$^{-1}$ bar$^{-1}$. The cycle of light-induced temperature-responsive gating performance can be repeated as the Fig. 3c shows without obvious decay. The photocontrol of

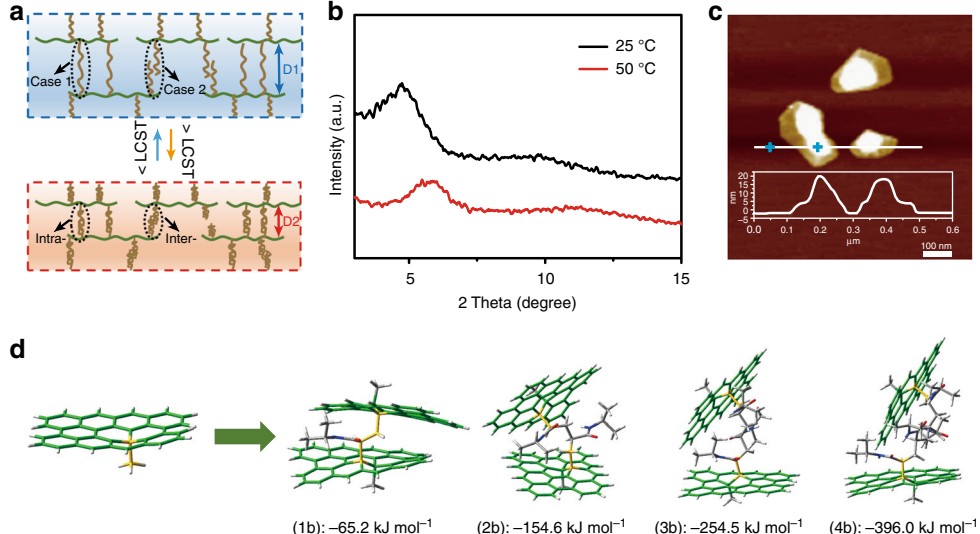

**Fig. 4** Mechanism of the negative temperature-response behavior of P-GOMs. **a** Schematic diagram of the channel size change of P-GOMs from D1 to D2 caused by the swelling or shrinking of PNIPAM chains covalent bound to GO sheets. Case 1: PNIPAM chains grow from one GO sheet and terminate at another GO sheet. Intramolecular hydrogen bonding is formed when the temperature is above its LCST. Case 2: PNIPAM chains grow from one GO sheet and terminate freely. Intermolecular hydrogen bonding is formed when the temperature is above its LCST. **b** The XRD patterns of P-GOMs at 25 °C and 50 °C. The right shift of the diffraction peak indicates the smaller lamellar distance at 50 °C than 25 °C. **c** The AFM image of P-GO. The height of cross-sectional profile proves that the P-GO sheets would like to stack together at 50 °C. **d** Optimized geometries of $[C_{36}H_{16}\text{-}CH_3]\bullet$ and the representative products of reactions (1b) $C_{36}H_{16}\text{-}CH_3\text{-}NIPAM\text{-}CH_3\text{-}C_{36}H_{16}$, (2b) $C_{36}H_{16}\text{-}CH_3\text{-}(NIPAM)_2\text{-}CH_3\text{-}C_{36}H_{16}$, (3b) $C_{36}H_{16}\text{-}CH_3\text{-}(NIPAM)_3\text{-}CH_3\text{-}C_{36}H_{16}$ and (4b) $C_{36}H_{16}\text{-}CH_3\text{-}(NIPAM)_4\text{-}CH_3\text{-}C_{36}H_{16}$. Others are shown in Supplementary Fig. 16. The graphene sheet is shown in green tubes, the PNIPAM as tubes (atomic color scheme: C, gray; N, blue; O, red; H, white). The newly formed bond is highlighted in yellow

water transport through P-GOMs may expand their applications such as in lab-on-a-chip settings[52].

**Mechanism of the negative temperature-response behavior of P-GOMs.** The unusual negative temperature-response behavior of P-GOMs, which is quite different from the GO/PNIPAM physical blending membranes[36], is due to the covalently bound interactions between PNIPAM chains and GO sheets (Supplementary Figs. 13 and 14, Supplementary Note 1). When synthesizing P-GO during the free-radical polymerization, there were two cases of the PNIPAM chains termination (except the self-polymerization of NIPAM): two ends of a PNIPAM chain termination by connecting two GO sheets and terminating freely at the bulk solution with only one end of the chain anchored to GO sheet. The AFM images of GO and P-GO at 25 °C presented in Supplementary Fig. 15 show that, both single- and multi-layered P-GO sheets exist because of the interactions of PNIPAM chains with GO sheets, whereas almost no stacked sheet is observed for GO sample. Figure 4a demonstrates that, for the first case, the covalently linked PNIPAM would pull two adjacent GO sheets closer by molecular chain contraction when the temperature is above its LCST. For the second case, part of the PNIPAM chains bound to adjacent GO sheets would form intermolecular hydrogen bonds when the temperature is above its LCST, which narrows the distance between two adjacent GO sheets because of the entanglement of PNIPAM chains[3]. Both these two cases result in a tunable GO lamellar space which play a dominant role in the negative temperature-response gating performance. This hypothesis was proved by XRD and AFM. For the P-GOMs in wet state, because PNIPAM was intercalated into the GO sheets to form a layered nanostructure, the diffraction peak of GO (Supplementary Fig. 10) disappeared and a new peak at $2\theta = 4.71°$ appeared. When the temperature increased to 50 °C, the diffraction peak of P-GOMs shifted right to

$2\theta = 5.91°$. The smaller lamellar distance at 50 °C proved the movement of GO sheets because of the shrinking of PNIPAM chains, resulting in a decreased water permeance. Besides, the entanglement of PNIPAM chains between two adjacent GO sheets also results in the decreased water permeance[53,54]. Furthermore, the temperature-response behavior of P-GO was characterized by AFM. The pre-heated P-GO dispersion (~ 50 °C) was spin-coated (2000 rpm) onto freshly cleaved mica to obtain the AFM sample[55]. As shown in Fig. 4c, the cross-sectional profile indicates the aggregation of P-GO sheets with heights of ~20 nm, implying the dense stacking of P-GO sheets at 50 °C.

In order to further understand the mechanism, quantum chemical calculations were performed using density functional theory to elucidate the reaction enthalpy for the formation of the graphene-PNIPAM. In radical polymerization, a polymer forms by the successive addition of radicals. The aim of this computational investigation is to establish whether the reaction of functionalized graphene with NIPAM monomers would terminate composite molecules freely in bulk solution or terminate at another functionalized graphene sheet. The polymerization reaction mechanism for the two possibilities along with the full computational details are given as Supplementary Note 2. Mylvaganam et al. found that the carbon adjacent to the $sp^3$ carbon is prone to be attacked by an ethylene moiety[56]. Owing to steric factors the graphene sheet is expected to react with NIPAM molecules on the opposite side of the methyl group. Supplementary Fig. 16 and Fig. 4d show the optimized geometries of these products in reactions (1a–4a) and (1b–4b) (Supplementary Note 2). The calculated reaction enthalpies for these two sets of reactions are given in Supplementary Table 1. An important result is that the first two reactions involving one functionalized graphene sheet (1a and 2a) are endothermic, whereas the corresponding reactions including two functionalized graphene sheets (1b and 2b) are exothermic. Specifically, the reaction enthalpies at 298 K ($\Delta H_{298}$) were + 50.9 (1a), + 20.8 (2a), −65.2

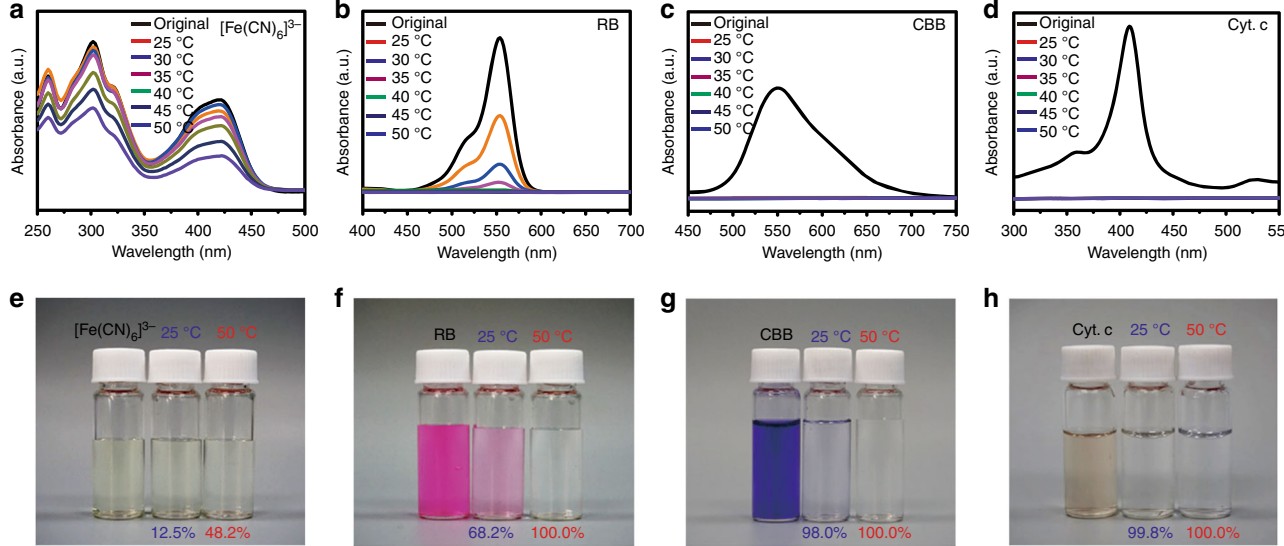

**Fig. 5** P-GOMs for single molecule separation. **a–d** The UV-vis absorption spectra before and after filtering $[Fe(CN)_6]^{3-}$ aqueous solutions, rhodamine B (RB) aqueous solutions, coomassie brilliant blue (CBB) aqueous solutions, and cytochrome c (Cyt. c) aqueous solutions from 25 °C to 50 °C. **e–h** The digital photos of original solutions and the filtrate obtained at 25 °C and 50 °C of $[Fe(CN)_6]^{3-}$, RB, CBB, and Cyt. c aqueous solutions

(1b), and −154.6 kJ mol$^{-1}$ (2b). For the consecutive reactions, three and four NIPAM monomers react with one or two functionalized graphene sheets. In this situation, all the reactions are exothermic. However, the reaction enthalpies for the reactions involving one functionalized graphene sheet were −158.3 (3a) and −262.7 kJ mol$^{-1}$ (4a), whereas the reactions involving two graphene sheets were much more exothermic with reaction enthalpies of −254.5 (3b) and −396.0 kJ mol$^{-1}$ (4b). The higher exothermicities of the later reactions could be partly attributed to the stronger van der Waals interactions between the graphene sheets and the side chain of the PNIPAM. These computational results indicate that the polymerization reaction between graphene and PNIPAM will prefer to be terminated by another functionalized graphene sheet as case (1) showed in Fig. 4a (Supplementary Note 3). We have shown that this is the case for 1–4 NIPAM molecules, however this trend is expected to continue for larger numbers of NIPAM molecules.

**Separation performance of the P-GOMs**. To validate the results of temperature tunable channel size of P-GOMs discussed above, we designed a separation experiment to test the lamellar space change at different temperatures by using small molecules with different sizes. Here we employed five ions/molecules, $Cu^{2+}$ (0.8 nm), $[Fe(CN)_6]^{3-}$ (0.9 × 0.9 nm), rhodamine B (RB, 1.8 × 1.4 nm), coomassie brilliant blue (CBB, 2.7 × 1.8 nm), and cytochrome c (Cyt. c, 2.5 × 2.5 × 3.7 nm)[22,29,57]. The rejection rates of the five ions/molecules through the P-GOMs from 25 °C to 50 °C were measured. The zeta potential results (Supplementary Fig. 17) demonstrate that the P-GOMs have few anionic sites for adsorption of the positively charged molecules and rejection of the negatively charged molecules because of the relatively low absolute value of the zeta potential. Thus size-dependent rejection rates nearly indicate the nanochannels size change of P-GOMs[58,59]. The results in Figs. 5 and 6a demonstrate that the nanochannel sizes of the P-GOMs gradually decrease with the increasing temperature (Supplementary Table 2). These results not only verified the results of quantum chemical calculations, but also demonstrated that the negative temperature-response P-GOMs could be promising for precise molecular separation (Supplementary Figs. 18 and 19, Supplementary Note 4).

Moreover, we suppose that the P-GOMs could realize gradient separation of multiple molecules with different sizes.

The multiple molecules separation schematic diagram is illustrated in Fig. 6b. The smallest molecule A is separated out at 50 °C, leaving the medium and largest molecules (B and C) in the retentate. Then the temperature changes to 25 °C to separate the medium size molecule from the large molecule. Thus, we can separate multiple molecules with different sizes using the P-GOMs by simple regulating the temperature. Subsequently, we designed a separation scenario using the mixed molecules solution, containing $Cu^{2+}$, RB, and Cyt. c (Fig. 6c, d). Figure 6e shows the UV-vis absorption spectra of filtrates obtained at 50 °C and 25 °C and retentate. For the $Cu^{2+}$, it only existed in the filtrate obtained at 50 °C according to the results of inductively coupled plasma optical emission spectrometry. In the filtrate obtained at 50 °C, there were trace RB and Cyt. c according to its UV-vis absorption spectrum. For RB, it enriched in the filtrate obtained at 25 °C. Therefore, only the largest Cyt. c remained in the retentate obtained at 25 °C. The results verified the selective separation performance of the P-GOMs to the mixed ions/molecules solution (Supplementary Fig. 20).

We have reported a smart temperature-response gating membrane composed of GO and PNIPAM for hydraulic permeability applications. The membranes have negative response coefficients with a gating ratio of ~7. What's more, the gating membranes could realize gradual small/medium/large molecule separation by simple step-wise tuning of the temperature. This study provides insight toward the design of smart gating membranes, showing promise for applications in fluid transport systems, microfluidic chip systems, and molecular separation devices.

## Methods
**Materials**. GO was prepared by a modified Hummer's method. Ethanol, 2,2'-Azobis(2-methylpropionitrile) (≥ 99.0%, recrystallization), N, N-dimethylformamide (DMF, ≥ 99.8%, anhydrous), copper (II) chloride (CuCl$_2$), potassium ferricyanide (K$_3$[Fe(CN)$_6$]), rhodamine B (RB), coomassie brilliant blue, cytochrome c (Cyt. c), and raffinose were all purchased from Aladdin Chemistry Co., Ltd. (Shanghai, China) and used as received. Maltopentaose and maltoheptaose (HPLC) were provided by Miragen (USA). NIPAM was provided by Sigma-Aldrich (Shanghai, China) and purified by recrystallization from n-hexane (≥ 99%, Aladdin). PNIPAM (M$_n$ ~ 20,000–40,000) was provided by Sigma-Aldrich.

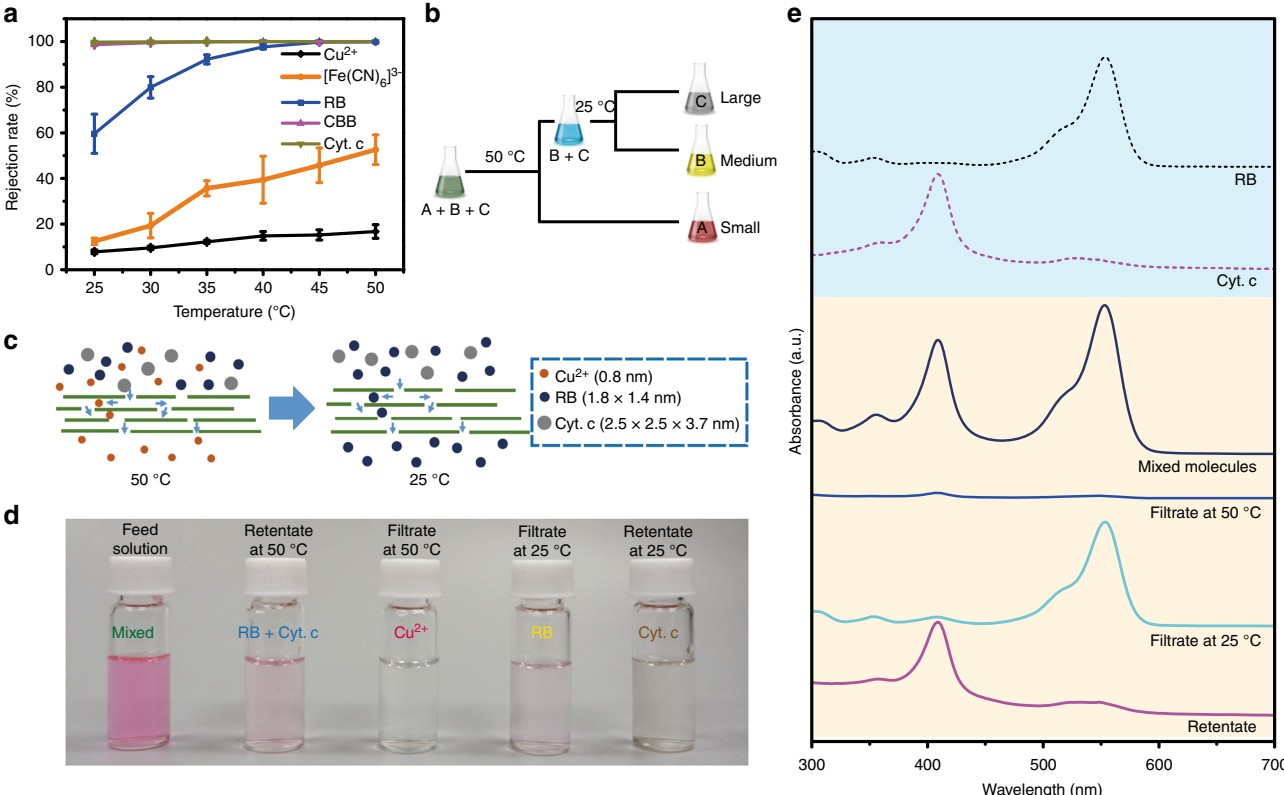

**Fig. 6** P-GOMs for multiple molecules separation. **a** Temperature-dependent rejection rates of P-GOMs to $Cu^{2+}$, $[Fe(CN)_6]^{3-}$, RB, CBB, and Cyt. c, implying smaller channel size of the P-GOMs with the increasing temperature. Error bars, s.d. ($n = 3$). **b** Schematic diagram of the separation mechanism of P-GOMs towards mixed molecules solution. **c** Separation of mixed molecules solution, containing $Cu^{2+}$, RB, and Cyt. c. **d** The digital photos of the mixed molecules feed solution, retentate at 50 °C, filtrate at 50 °C, filtrate at 25 °C, and retentate at 25 °C. **e** The UV-vis absorption spectra of pure RB aqueous solution, Cyt. c aqueous solution, mixed molecules solution, filtrate obtained at 50 °C and 25 °C, and retentate at 25 °C

**Preparation of P-GO dispersion.** The P-GO were synthesized by free-radical polymerization. Typically, GO (0.05 g) was first fully dispersed in DMF (50 ml) by stirring for 1 h and ultrasonic treatment for 2 h. Afterwards, NIPAM (50 mmol, 5.659 g) and 2,2'-Azobis(2-methylpropionitrile) (0.056 g) were added to the dispersion, respectively. The mixture was subsequently transferred to a 100 ml Schlenk flask followed by stirring for 30 mins. Next, the reagent mixture was subjected to three freeze-pump-thaw cycles and was then placed in a 65 °C oil bath. The reaction was terminated after 12 h by immersing the flask into liquid nitrogen. The obtained solution was diluted with 250 ml DI water and transferred into centrifuge tubes followed by 1 h centrifugation at the speed of 10,000 r min$^{-1}$ (10,744 g). The precipitates located in the bottom of centrifuge tubes after centrifugation were collected and washed three times each with 300 ml DI water and 300 ml ethanol. To make sure NIPAM monomer and the PNIPAM not bound to GO were completely separated from the P-GO, the precipitates were purified through dialysis in DI water for 3 days using dialysis bag (MW cutoff: 30,000). Then the P-GO were obtained after freezing dry of the dispersion for 2 days with a freezing dryer. A brownish transparent, stable P-GO dispersion (0.1 mg ml$^{-1}$) was obtained by adding 0.01 g of the as-prepared P-GO in 100 ml DI water and stirring for 2 h. Note, no ultrasound treatment was needed to obtain the well-dispersed P-GO aqueous suspension during the whole synthesis process except dispersing GO into DMF at first.

**Preparation of GO dispersion.** GO (0.009 g) was added into 100 ml DI water followed by stirring for 1 h and ultrasonic treatment for 2 h to obtain a well-dispersed GO aqueous dispersion (0.09 mg ml$^{-1}$).

**Preparation of GO blending with PNIPAM dispersion.** PNIPAM (0.021 g) was added into 100 ml GO aqueous dispersion (0.09 mg ml$^{-1}$) and stirred for 6 h to obtain the homogeneous GO blending with PNIPAM dispersion.

**Fabrication of P-GOMs, GOMs, and GO blending with PNIPAM membranes.** For the preparation of a P-GOM, 6 ml (except discussing as variable) P-GO aqueous dispersion (0.1 mg ml$^{-1}$) was filtered through a CA membrane by pressure-driven filtration (Millipore 8010, Millipore, USA) with a filter area of 2.83 cm$^2$ under the pressure of 100 kPa. Once no P-GO dispersion left above the formed membrane, 10 ml DI water was immediately added to the filtration cell to keep the

P-GOM in a wet state. The mass of GO in a P-GOM was ~0.00018 g based on the calculation from the TGA and differential thermogravimetrie curves as described in Supplementary Note 1. Thus, the procedure was basically the same as the P-GOM for the fabrication of GOM or GO blending with PNIPAM membrane, except using 2 ml GO dispersion (0.09 mg ml$^{-1}$) or 2 ml GO blending with PNIPAM dispersion (containing 0.00018 g GO) instead of the P-GO dispersion. The GO mass in three kinds of membranes were all fixed at ~0.18 mg. The average values were obtained by measuring three independent samples for each. What is more, in order to keep the GOM stable in water, a negative pressure was applied to the pressure-driven filtration when added 10 ml DI water above the GOM until putting pressure on the membrane.

**Permeability performances of P-GOMs, GOMs and GO blending with PNI-PAM membranes.** For the temperature-responsive hydraulic permeability experiments, a water bath was used to regulate the temperature of the membrane. The filtration device contained the formed membrane and 10 ml DI water was put into the water bath with desired temperature. After standing for 30 mins, 100 kPa pressure generated by $N_2$ was applied to the membrane. The weight of water penetrated through the membrane was recorded every 30 mins, and the water permeance was calculated using the value which kept constant for >90 mins.

For the photo-responsive hydraulic permeability experiments, a vacuum filtration system combined with an 808 nm NIR laser was used.

**Molecular separation experiments of P-GOMs.** While for the molecule separation experiments, the aqueous solutions of 100 mg l$^{-1}$ CuCl$_2$, 40 mg l$^{-1}$ K$_3$[Fe(CN)$_6$], 4 mg l$^{-1}$ RB (Supplementary Fig. 21), 40 mg l$^{-1}$ coomassie brilliant blue (Supplementary Fig. 22), 100 mg l$^{-1}$ Cyt. c, 200 mg l$^{-1}$ raffinose, 200 mg l$^{-1}$ maltopentaose and 200 mg l$^{-1}$ maltoheptaose were used instead of DI water, and the water bath contained the filtration device with a stir bar was placed on a magnetic heated stirrer in case of concentration polarization. After certain time, the filtrates were collected. The rejection rate ($R$) was calculated using the following equation:

$$R = 1 - (c_p/c_o) \times 100\% \qquad (2)$$

where $c_p$ and $c_o$ are the concentrations of the molecule in the permeates and original solutions.

For the mixed molecules separation experiments, 5 ml aqueous solution contained 100 mg l$^{-1}$ Cu$^{2+}$, 4 mg l$^{-1}$ RB, and 100 mg l$^{-1}$ Cyt. c was used as the feed solution. First, the filtration device was put in the 50 °C water bath, after standing for 30 mins, 100 kPa pressure was applied to the membrane. After collecting ~4.5 ml filtrate, 4.5 ml DI water was added into the filtration device, followed by the second separation at 50 °C. The temperature was altered to 25 °C until no Cu$^{2+}$ could be detected in the filtrate at 50 °C. Like the separation process at 50 °C, after several separations at 25 °C, the filtrate contained only RB could be obtained. The retentate contained only Cyt. c could be obtained as well in the end.

**Computational details**. To simplify the calculations, we used a methyl radical (CH$_3$•) to mimic the experimentally carbon related radical (namely CN(CH$_3$)$_2$C•) as the initiator. A graphene sheet containing 36 carbon atoms, C$_{36}$H$_{16}$, was used to study the polymerization reaction instead of GO used in experiment because the polymer chains were grafted to graphitic surface utilizing the double bonds of surfaces in the free-radical polymerization[42]. Using graphene sheet did not influence the results, however, substantially saved computational cost. The initiation, propagation, and termination steps of a free-radical polymerization reaction were carried out by using NIPAM as a monomer and CH$_3$• as the initiator. The reaction of a methyl radical with graphene and the subsequent reaction of the newly formed graphene-methyl radical with NIPAM were examined for the initiation step. The subsequent reaction of graphene-methyl- NIPAM radical with another NIPAM monomer was examined for the propagation step. The termination step was examined by considering the reaction of the resultant radical with another methyl functionalized graphene sheet. For every step, geometries of the reactants and products were optimized using density functional theory method with a hybrid functional B3LYP[60–62] and a 6–31 G(d, p) basis set. Empirical D3 dispersion corrections are included using the Becke–Johnson damping potential as recommended in Ref.[63] (denoted by the suffix-D3). Bulk solvent effects of DMF ($\varepsilon = 37.2$) have been included using charge-density based SMD continuum solvation model at the same level of theory[64]. All calculations were performed by using Gaussian 09 package[65].

**Characterization**. Digital photographs were shot by a digital camera. Scanning electron microscope images were obtained on a scanning electron microscopy (JSM-7500F, JEOL, Japan) equipped with a energy dispersive spectroscopy detector. AFM images were carried out on a Bruker Dimension Icon (Japan). Diluted GO or P-GO aqueous dispersion was spin-coated (2000 rpm) on freshly cleaved mica and dried at room temperature to make the AFM samples. Thermographic images were obtained with a FLIR TG165 imaging IR thermometer (FLIR Systems, Inc., OR, USA). The values of contact angle were measured on an OCA20 machine (Dataphysics, Germany) at room temperature. ATR-FTIR spectra were measured by an infrared microspectrometer (Thermo Scientific Nicolet iN10, USA). XPS characterizations were performed by ESCALAB 250Xi (Thermo Fisher Scientific, USA) equipped with a monochromatic Al Kα X-ray source. Raman spectroscopy measurements were performed using a Lab-RAM HR800 (Horiba JobinYvon, France) with an incident laser of 514 nm wavelength. The XRD analyses were carried out by an X-ray diffractometer (XRD-6000, Shimadzu, Japan), using Cu Kα radiation ($\lambda = 0.15418$ nm). The $^1$H nuclear magnetic resonance spectra were obtained on a Bruker Avance III 700 MHz NMR spectrometer (Bruker Biospin, Rheinstetten, Germany). TGA was performed on a thermogravimetric analyzer (STA449F3, Netzsch, Germany), with a temperature increase rate of 10 K min$^{-1}$ under nitrogen from 25 °C to 750 °C. The thicknesses of P-GOMs were measured by Bruker DektakXT Stylus Profiler (Germany). The concentrations of Cu$^{2+}$ in the solution were analyzed using inductively coupled plasma optical emission spectrometry (ICP-OES, Optima 5300DV, PerkinElmer, USA). The concentrations of raffinose, maltopentaose and maltoheptaose in the solution were analyzed by ion chromatography equipped with a CarboPac PA200 3 × 250 column. UV-vis absorption spectra were obtained with a Shimadzu UV-3600 spectrometer (Japan) equipped with a thermal controller. Zeta potentials were analyzed by DelsaNano C Zeta Potential Analyzer (Beckman Coulter Inc., USA). Tensile mechanical properties were measured at room temperature (25 °C) via a Shimadzu AGS-X Tester (Japan) with a 20 N load cell at a loading rate of 1 mm min$^{-1}$. The samples were cut into strips with a width of 3 mm and length of 10 mm, and the thicknesses of all samples were calculated by scanning electron microscope thickness measuring instrument.

**Data availability**. All data generated or analyzed during this study are present in the main text and the Supplementary Information. Additional data are available from the authors on reasonable request.

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

## Acknowledgements

We are grateful for the financial support from the National Natural Science Foundation of China (21774005, 21433012, 21374001, 21222309), the National Youth Talent Support Program, the Program for New Century Excellent Talents in University of China, the Fundamental Research Funds for the Central Universities, the China Scholarship Council (201506025110) and the National Instrumentation Program (2013YQ120355). The work at Harvard was supported by the NSF (DMR-1310266) and the Harvard MRSEC (DMR-1420570).

## Author contributions

Y.Z. conceived the study and designed the experiments. J.C.L. performed most of the experiments. L.J.Y and A.K. performed the density functional theory calculations. W.L., W.X.Z., and Q.F.C. provided some valuable scientific suggestions about the temperature-responsive properties of the P-GOMs. F.Y.G., and L.L.H. performed the tensile experiments and analyzed the data. J.C.L. and N.W. wrote the text. Y.Z., N.W., L.J., and D.A.W revised the paper.

## Additional information

**Competing interests:** The authors declare no competing financial interests.

9