## [Peer Review File · Nature Communications]

Reviewers' comments:

Reviewer #1 (Remarks to the Author):

The authors reported a negative temperature-response gating membrane composed of GO and PNIPAM. They not only show the tunable water gating and precise small molecules separation of the membrane but also explains the mechanism of the temperature-responsive behaviour. The results are interesting.

However, there are some issues that need to be addressed.

1. Ref. 36 also involves the use of chemically reduced GO and PNIPAM for smart membrane application. The authors should clearly present how this work is different from Ref. 36

2. In line 143, page 8, authors indicate there are "More characterization including Raman spectra, and ^1H nuclear magnetic resonance (^1H NMR)" but do not have the explanation of the data. They should give the explanation either here or in supplementary information.

3. In line 173, page 10, authors show "This hypothesis was supported by the results of the XRD, water contact angle, XPS, and Raman characterizations (Supplementary Fig. S10)". But there are no XPS results in Supplementary Fig. S10.

4. In line 260, page 14, authors XRD peak positions of P-GOMs at 25 °C and 50 °C. By calculation, $2\theta = 5.43^\circ$ corresponds to a layer distance of 1.627 nm, $2\theta = 5.59^\circ$ corresponds to a layer distance of 1.581 nm, the difference between them is 0.046 nm. I think this layer distance is so small that cannot explain the large water flux difference. This may be not real layer distance because samples were dried for XRD. I would strongly recommend the authors do the XRD of the membranes in wet state.

5. In line 266, page 15, "Fig. 4b" should be "Fig. 4c".

6. In line 283, page 16, "c)" should be "d)".

7. In line 301, page 17, "Fig. 4c" should be "Fig. 4d".

Reviewer #2 (Remarks to the Author):

This is an interesting paper, authors reports the modulation of water flux in GO based membrane by polymerization of a temperature-responsive polymer with GO sheets. I think it opens a new route on controlling the laminar structure and modulating the mass transport through GO membrane. The idea and experimental data are fairly solid. However, on the mechanism understanding, I have several comments that need to be addressed.

1. Authors explained the gating of water transport by shrinkage of channel inside the GO membrane, however, from the Fig. 4c, when the membrane was heated to 50°C, one can only see <0.5 degree shifting in XRD spectrum, this means there is only ~0.2 nm shrinking on the channel, the little change should not be enough to explain either the 7 times decrease of water flux(if calculated by Hagen-Poiseuille equation, that is equation 1 in manuscript), or the sieving properties(RB show 60% rejection at 25 °C, which increase to 90% at 50 °C). I noticed the XRD was performed with relative humidity of 40%, I think, in fact, this could not be used as an effective evidence to show the channel width dependence on temperature. XRD measurements of samples immersed in liquid water, or at least 100% humidity, which reflects the real filtration conditions, to see if there is a large enough shrinkage would be a appropriate evidence. If after the XRD measurement, the change of channels is still not enough to explain the experiment results, then a proper explanation should be made to gain the insights on the

mechanism.

2. The thickness of individual P-GOM sheets is ~5 nm, the thickness of PNIPAM layer is then estimated to be 4 nm (page 7, line 122-125). This estimation is based on there is only one layer of GO encapsulated by PNIPAM, however, AFM could not prove there is only one layer GO sandwiched/wrapped by PNIPAM. In particular, GO is prone to agglomerate during the proposed process, as the precursor- NIPAM is positively charged, then, the GO sheets probably are multi-layer stack. Authors need a justification on this statement.

3. One inconsistency data. the rejection of $[\text{Fe}(\text{CN})_6]$ is written as 38.8% in Fig. 5e, however, from Fig. 5i, it is 10%-20%.

The Y axis labels in Fig. 3C, from bottom to up, 0,2,4,6,8,(1)0,(1)2,(1)4, similar issue could be found in Fig 5i.

one Typo: Fig. 5a-d, Absortance.

Citations or a justification is needed on the source of size of ions/molecules (Page 18).

Reviewer #3 (Remarks to the Author):

This manuscript reports the preparation of a PNIPAM-GO membrane where thermal properties of PNIPAM allow the spacing between the GO sheets to respond with respect to temperature. The work combines experimental and simulation evidence to support the proposed mechanism. When the temperature exceeds a certain value, the GO sheets collapse and shutdown transport of water. Intermediate temperatures can be used to do selective separations in the several nm to order 10 nm molecular size range. The results seem reasonable and are accompanied by a fair amount of hype. It would be helpful to back the hype up with comparisons to existing membranes that are used to separate the types of molecules that are considered in this work. Instead, the manuscript largely stays away from any quantitative comparison, preferring to rely on language to convince the reader that the materials are particularly special (from an application perspective) - no doubt, though, that the physical mechanism is interesting.

For example, it would be useful to report permeance and/or permeability data for transport rates as opposed to just flux. That would help the reader to compare these materials to other membranes where, particularly permeance units, are commonly reported.

Presumably the materials aren't useful for desalination, or else data with sodium would have been used. One is left to wonder what the actual separation resolution really is for these materials. The 'hype' would be more justified by doing separations between ions, for example, as opposed to the Cu^{2+} and RB cutoff that is reported (i.e., a hydrated 'big' ion versus an organic dye). They authors should at least report the resolution of separations that can reasonably be achieved for dye molecules versus ions using other membrane approaches.

Additionally, no rationale is given for the low concentrations that were used in the experiments. One is left to wonder whether those conditions are really relevant / how the material would perform if challenged with a more concentrated feed solution.

As an aside, it seems unlikely that highly confined water molecules in the nano-channels would behave like a bulk viscous flow (such that the Hagen-Poiseuille equation would be valid). It seems as though a more molecular approach to analyzing transport would be appropriate compared to a continuum approach.

We are greatly encouraged that all three referees affirm our work “interesting”. The comments are invaluable and very helpful for improving our paper, as well as the important guiding significance to our research. We have evaluated comments carefully and made revisions according to these suggestions. Revised portions are marked in yellow in the paper. The main revisions in the paper and the responses to the reviewers’ comments are as following:

Responses to Reviewers’ comments:

Reviewer #1:

The authors reported a negative temperature-response gating membrane composed of GO and PNIPAM. They not only show the tunable water gating and precise small molecules separation of the membrane but also explain the mechanism of the temperature-responsive behaviour. The results are interesting.

Response:

We appreciate reviewer’s very encouraging comment.

Comment 1:

Ref. 36 also involves the use of chemically reduced GO and PNIPAM for smart membrane application. The authors should clearly present how this work is different from Ref. 36

Response 1:

Thanks for the comment. Our work is quite different to Ref. 36 in several aspects. Firstly, the fabrication methods were different. Our PNIPAM grafted GO membrane was composed of PNIPAM covalently bound graphene oxide *via* free radical polymerization, while the membrane in Ref. 36 was fabricated *via* physical blending reduced GO and PNIPAM. Secondly, the chemically reduced GO and PNIPAM reported by Ref. 36 were connected by supramolecular interaction, which showed a positive temperature-response gating behavior. The channel size increased when temperature was above the LCST of PNIPAM. Therefore, the water flux of membrane increased with the increasing temperature and the rejection rate for molecules decreased. In our work, the membrane was composed of PNIPAM covalently bound GO *via* free radical polymerization. The covalently bound interactions between PNIPAM chains and GO endowed the membrane negative temperature-response gating behavior opposite to Ref. 36. The channel size decreased because of the shrinkage of PNIPAM chains when the temperature was above its LCST, leading to the decreased flux and increased molecule rejection rate

(Figure 4a). Thirdly, the channel size tunable ability of our PNIPAM covalently bound GO membrane endowed it realizing gradient separation of multiple molecules with different sizes by regulating the temperature, which was not achieved before.

We have added the comparison in the revised manuscript (page 14, line 241) and added the difference between Ref. 36 and our work in the revised Supplementary Information (page 9, line 95)

Comment 2:

In line 143, page 8, authors indicate there are “More characterization including Raman spectra, and ^1H nuclear magnetic resonance (^1H NMR)” but do not have the explanation of the data. They should give the explanation either here or in supplementary information.

Response 2:

Thanks for referee’s kind advice. We have added the detailed explanations about the Raman spectra (page 5, line 51) and ^1H nuclear magnetic resonance (^1H NMR) (page 6, line 61) in the revised Supplementary Information.

Comment 3:

In line 173, page 10, authors show “This hypothesis was supported by the results of the XRD, water contact angle, XPS, and Raman characterizations (Supplementary Fig. S10)”. But there are no XPS results in Supplementary Fig. S10.

Response 3:

We are sorry for the omission. We have added the XPS spectra (Fig. R1) and explanation in the revised Supplementary Information (Supplementary Fig. S10, page 8).

Fig. R1. The survey scan XPS spectra of GO and GO-12 show the C/O atomic ratio increased from 2.06 (GO) to 2.36 (GO-12), demonstrating the reduction of GO.

Comment 4:

In line 260, page 14, authors XRD peak positions of P-GOMs at 25°C and 50°C. By calculation, $2\theta = 5.43^\circ$ corresponds to a layer distance of 1.627 nm, $2\theta = 5.59^\circ$ corresponds to a layer distance of 1.581 nm, the difference between them is 0.046 nm. I think this layer distance is so small that cannot explain the large water flux difference. This may be not real layer distance because samples were dried for XRD. I would strongly recommend the authors do the XRD of the membranes in wet state.

Response 4:

We appreciate this very helpful suggestion. In the initial manuscript, we performed the XRD characterizations of P-GOMs in dry state, the diffraction peak of P-GO shifted from $2\theta = 5.43^\circ$ to $2\theta = 5.59^\circ$. However, this could not fully reflect the real lamella distance change in aqueous environment as the referee indicated. Following the reviewer's good suggestion, we have performed the XRD characterization of the membranes in wet state. The diffraction peak shifts from $2\theta = 4.71^\circ$ to $2\theta = 5.91^\circ$ when the temperature increased from 25°C to 50°C, demonstrating the d-spacing of P-GOMs changes from 1.87 nm to 1.49 nm. The smaller d-spacing of P-GOMs at 50°C, undoubtedly, will decrease the water flux through the P-GOMs. Besides, the entanglement of PNIPAM chains between two adjacent GO sheets also leads to the decrease of water flux, because the entangled PNIPAM chains will hinder the water transport by decreasing the effective channel size [Chu, L. Y., Niitsuma,

T., Yamaguchi, T. & Nakao, S. i. Thermoresponsive transport through porous membranes with grafted PNIPAM gates. *AIChE J.* **49**, 896-909 (2003); Li, Y. et al. Thermoresponsive gating characteristics of poly (N-isopropylacrylamide)-grafted porous poly (vinylidene fluoride) membranes. *Ind. Eng. Chem. Res.* **43**, 2643-2649 (2004)]. As a result, the large water flux difference of P-GOMs at 25°C and 50°C occurred under the synergistic effects of smaller d-spacing and PNIPAM chains entanglement.

In the revised manuscript, we replaced the original dry state XRD spectra in Fig. 4b (page 16) with the wet state XRD spectra below (Fig. R2). And we added the explanation of the large water flux difference in the revised manuscript (page 15, line 262).

Fig. R2 The XRD patterns of P-GOMs at 25°C and 50°C. The right shift of the diffraction peak indicates the smaller lamellar distance at 50°C than 25°C.

Comment 5:

In line 266, page 15, “Fig. 4b” should be “Fig. 4c”. In line 283, page 16, “c)” should be “d)”. In line 301, page 17, “Fig. 4c” should be “Fig. 4d”.

Response 5:

Great appreciation for the referee’s rigorous and responsible comment. We have revised “Fig. 4b” to “Fig. 4c”. (page 15, line 266), “c)” to “d)” (page 16, line 284), and “Fig. 4c” to “Fig. 4d” (page 17, line 302).

Reviewer #2:

This is an interesting paper, authors report the modulation of water flux in GO based membrane by polymerization of a temperature-responsive polymer with GO sheets. I think it opens a new route on controlling the laminar structure and modulating the mass transport through GO membrane. The idea and experimental data are fairly solid. However, on the mechanism understanding, I have several comments that need to be addressed.

Response:

We appreciate very much for the referee's highly affirmative comment.

Comment 1:

Authors explained the gating of water transport by shrinkage of channel inside the GO membrane, however, from the Fig. 4c, when the membrane was heated to 50°C, one can only see <0.5 degree shifting in XRD spectrum, this means there is only ~0.2 nm shrinking on the channel, the little change should not be enough to explain either the 7 times decrease of water flux (if calculated by Hagen-Poiseuille equation, that is equation 1 in manuscript), or the sieving properties (RB show 60% rejection at 25°C, which increase to 90% at 50°C). I noticed the XRD was performed with relative humidity of 40%, I think, in fact, this could not be used as an effective evidence to show the channel width dependence on temperature. XRD measurements of samples immersed in liquid water, or at least 100% humidity, which reflects the real filtration conditions, to see if there is a large enough shrinkage would be an appropriate evidence.

If after the XRD measurement, the change of channels is still not enough to explain the experiment results, then a proper explanation should be made to gain the insights on the mechanism.

Response 1:

Thanks for referee's helpful comment. In the initial manuscript, we performed the XRD characterizations of P-GOMs in dry state, the diffraction peak of P-GO shifted from $2\theta = 5.43^\circ$ to $2\theta = 5.59^\circ$. As the referee indicated, this could not fully reflect the real lamella distance change in aqueous environment. Following the reviewer's suggestion, we have performed the XRD characterization of the membranes in wet state. The diffraction peak shifts from $2\theta = 4.71^\circ$ to $2\theta = 5.91^\circ$ when the temperature increased from 25°C to 50°C, demonstrating the d-spacing of P-GOMs changes from 1.87 nm to 1.49 nm based on Bragg equation:

$$2d \sin\theta = n\lambda$$

where $n = 1$, and $\lambda = 0.154$ nm.

Fig. R3 The XRD patterns of P-GOMs at 25°C and 50°C. The right shift of the diffraction peak indicates the smaller lamellar distance at 50°C than 25°C.

The smaller d-spacing of P-GOMs at 50°C will decrease the water flux through the P-GOMs. Besides, the entanglement of PNIPAM chains between two adjacent GO sheets also results in the decrease of water flux because the entangled PNIPAM chains will hinder the water transport by decreasing the effective channel size. [Chu, L. Y., Niitsuma, T., Yamaguchi, T. & Nakao, S. i. Thermoresponsive transport through porous membranes with grafted PNIPAM gates. *AIChE J.* **49**, 896-909 (2003); Li, Y. et al. Thermoresponsive gating characteristics of poly (N-isopropylacrylamide)-grafted porous poly (vinylidene fluoride) membranes. *Ind. Eng. Chem. Res.* **43**, 2643-2649 (2004)]. Therefore, the large water flux difference of P-GOMs at 25°C and 50°C occurred under the synergistic effects of smaller d-spacing and PNIPAM chains entanglement. Because some parameters in Hagen-Poiseuille equation were uncertain such as surface porosity and tortuosity at 25°C and 50°C, we used Hagen-Poiseuille equation to qualitatively describe the thickness effect of P-GOMs on the water flux in the manuscript. Thanks again for this valuable recommendation, which helps us get better understanding of our results.

In the revised manuscript, we replaced the XRD spectra in Fig. 4b (page 16) with the XRD spectra below (Fig. R3). And we added the explanation of the large water flux difference in the revised manuscript (page 15, line 262).

Comment 2:

The thickness of individual P-GOM sheets is ~5 nm, the thickness of PNIPAM layer is then estimated to be 4 nm (page 7, line 122-125). This estimation is based on there is only one layer of GO encapsulated by PNIPAM, however, AFM could not prove there is only one layer GO sandwiched/wrapped by PNIPAM. In particular, GO is prone to agglomerate during the proposed process, as the precursor- NIPAM is positively charged, then, the GO sheets probably are multi-layer stack. Authors need a justification on this statement.

Response 2:

We appreciate this important comment. AFM has been demonstrated to be a powerful tool for the thickness measurement of graphene, GO sheets, and other 2D molecular brushes [Kan, L., Xu, Z. & Gao, C. General avenue to individually dispersed graphene oxide-based two-dimensional molecular brushes by free radical polymerization. *Macromolecules* **44**, 444-452 (2010)]. To identify the real thickness of an individual P-GO sheet, we further carried out AFM characterizations by spin-coating the low concentration P-GO aqueous dispersion (0.002 mg/mL) as shown in Fig. R4. It is found that the smallest thickness of well-dispersed sheets of P-GO is at basically same height of about 5 nm. Thus, we deduce that the thickness of an individual P-GO sheet is about 5 nm.

Fig. R4 The AFM image of P-GO sheets, the thicknesses of two dispersed sheets are about 5 nm.

Comment 3:

One inconsistency data. the rejection of $[\text{Fe}(\text{CN})_6]$ is written as 38.8% in Fig. 5e, however, from Fig. 5i, it is 10%-20%.

The Y axis labels in Fig. 3C, from bottom to up, 0,2,4,6,8,(1)0,(1)2,(1)4, similar issue could be found in Fig 5i.

one Typo: Fig. 5a-d, Absortance.

Response 3:

Thanks for your helpful comment. We mistook the rejection rate at 35°C (38.8% in Fig. 5i) as 25°C. We have changed “38.8%” to “12.5%” in the revised manuscript (Fig. 5e, page 21) and Supplementary Information (Supplementary Table S2, page 21). Accordingly revisions have been made in Fig. 3c (page 13) and Fig. 5i (page 21). We have amended “Absortance” to “Absorbance” in the revised manuscript and Supplementary Information. Many thanks for the conscientious referee.

Comment 4:

Citations or a justification is needed on the source of size of ions/molecules (Page 18).

Response 4:

Thanks for this helpful suggestion. We have added the references in the revised manuscript to introduce the source of sizes of ions/molecules we used. (Refs. 22, 29 and 57, page 18, line 326)

Reviewer #3:

Comment 1:

This manuscript reports the preparation of a PNIPAM-GO membrane where thermal properties of PNIPAM allow the spacing between the GO sheets to respond with respect to temperature. The work combines experimental and simulation evidence to support the proposed mechanism. When the temperature exceeds a certain value, the GO sheets collapse and shutdown transport of water. Intermediate temperatures can be used to do selective separations in the several nm to order 10 nm molecular size range. The results seem reasonable and are accompanied by a fair amount of hype.

It would be helpful to back the hype up with comparisons to existing membranes that are used to separate the types of molecules that are considered in this work. Instead, the manuscript largely stays away from any quantitative comparison, preferring to rely on language to convince the reader that the materials are particularly special (from an application perspective) - no doubt, though, that the physical mechanism is interesting.

For example, it would be useful to report permeance and/or permeability data for transport rates as opposed to just flux. That would help the reader to compare these materials to other membranes where, particularly permeance units, are commonly reported.

Response 1:

We appreciate the referee's high evaluation of our work "no doubt, though, that the physical mechanism is interesting". In our work, a pressure of 1 bar was applied to the pressure-driven filtration (described in the Methods section of manuscript, page 25, line 446). Thus the flux data and permeance data are equal in number. We have converted the flux data into permeance data in the revised manuscript and Supplementary Information. And we also added the permeance data of P-GOMs for molecules at 25°C and 50°C in Supplementary Table S2 (revised Supplementary Information, page 21). In this work, the thickness of P-GOMs is fixed at 1.1 μm in order to achieve a larger temperature-response gating ratio (Fig. 3d in manuscript) and realize the rejection rate of almost 100% for RB at 50°C, thus improving the precision of mixed molecules gradient separation. However, separation membranes exhibit a trade-off between permeability and selectivity [Park, H. B., Kamcev, J., Robeson, L. M., Elimelech, M. & Freeman, B. D. Maximizing the right stuff: The trade-off between membrane permeability and selectivity. *Science* **356**, eaab0530 (2017)]. Therefore, we can improve the permeance by sacrificing a little selectivity of P-GOMs by using a thinner membrane. In Fig. 3d of the manuscript, we studied the effect of membrane thickness on temperature-response water gating ratio. We selected 1.1 μm membrane for water gating because it showed a good balance between the permeance and gating ratio. Therefore, the subsequent experiments also used such 1.1 μm membrane for molecular separation. If we decreased the membrane thickness to 0.8 μm , the RB rejection rate slightly decreased from near 100% (1.1 μm) to 98.4% as shown in Fig. R5. But its permeance significantly increased from 2.43 to 20.10 $\text{L m}^{-2} \text{h}^{-1} \text{bar}^{-1}$. According to the rejection rates for RB, the P-GOMs belong to nanofiltration-ranged separation membranes [Wang, X.-L., Shang, W.-J., Wang, D.-X., Wu, L. & Tu, C.-H. Characterization and applications of nanofiltration membranes: State of the art. *Desalination* **236**, 316-326 (2009)]. The permeance of 0.8 μm P-GOMs we prepared is two to four times higher than commercial nanofiltration membranes of about 5-10 $\text{L m}^{-2} \text{h}^{-1} \text{bar}^{-1}$ [Agenson, K. O., Oh, J.-I. & Urase, T. Retention of a wide variety of organic pollutants by different nanofiltration/reverse osmosis membranes: controlling parameters of process. *J. Membr. Sci.* **225**, 91-103 (2003); Boussu, K. et al. Characterization of polymeric nanofiltration membranes for systematic analysis of membrane performance. *J. Membr. Sci.* **278**, 418-427 (2006); Ang, H. & Hong, L. Polycationic Polymer-Regulated Assembling of 2D MOF Nanosheets for High-Performance Nanofiltration. *ACS Appl. Mater. Inter.* **9**, 28079-28088 (2017); Abozar, A. et al. Large-area graphene-based nanofiltration membranes by shear alignment of discotic nematic liquid crystals of graphene oxide. *Nat. Commun.* **7**, 10891 (2016)].

In response to these comments, we have made revisions in the revised manuscript (page 19, line 336) and Supplementary Information (page 14, line 143).

Fig. R5 The separation performance of 0.8 μm P-GOMs. a) The UV-vis absorption spectra before and after filtering RB from 25°C to 50°C using the 0.8 μm P-GOMs. b) Temperature dependent permeance and rejection rate of the 0.8 μm P-GOMs for RB.

Comment 2:

Presumably the materials aren't useful for desalination, or else data with sodium would have been used. One is left to wonder what the actual separation resolution really is for these materials. The 'hype' would be more justified by doing separations between ions, for example, as opposed to the Cu²⁺ and RB cutoff that is reported (i.e., a hydrated 'big' ion versus an organic dye). They authors should at least report the resolution of separations that can reasonably be achieved for dye molecules versus ions using other membrane approaches.

Response 2:

Thanks for referee's helpful comment. In the manuscript, the reason why we used Cu²⁺ and RB, two ion/molecule with big size difference, was to better describe the concept of temperature-tuned gradient separation which does not need to change membrane when separating mixed molecules. P-GOMs demonstrated large rejection rate difference for RB at 25°C and 50°C (RB could pass through the P-GOMs at 25°C, while could not at 50°C), and the rejection rate for Cu²⁺ was only 16.7% at 50°C. Therefore, we could effectively separate the small Cu²⁺ from the mixed ions/molecules solution at 50°C. Then the middle size RB could be separated at 25°C. Besides, we further chose three kinds of saccharides, raffinose (C₁₈H₃₂O₁₆), maltopentaose (C₃₀H₅₂O₂₆) and maltoheptaose (C₄₂H₇₂O₃₆), to perform the separation experiments. Raffinose, maltopentaose and maltoheptaose are non-ionic solutes with hydrated diameter of 1.32 nm, 1.52 nm and 1.96 nm, respectively [Schultz, S. G. & Solomon, A. Determination of the effective hydrodynamic radii of small molecules by viscometry. *J. Gen. Physiol.* **44**, 1189-1199 (1961)]. Firstly, the single-solute aqueous solutions of raffinose, maltopentaose and maltoheptaose with the concentration of 200 mg/L were used as feed solutions. The concentrations of raffinose, maltopentaose and maltoheptaose were measured by ion

chromatography (IC). As shown in Fig. R6, the P-GOMs showed rejection rates of 13.6% for raffinose, 34.3% for maltopentaose and 48.6% for maltoheptaose at 25°C. When the temperature was 50°C, the rejection rates increased to 35.3% for raffinose, 68.4% for maltopentaose and 94.3% for maltoheptaose, respectively. At 50°C, since the P-GOMs demonstrated a relatively high rejection rate for maltoheptaose and a low rejection rate for raffinose, we supposed that we could separate raffinose and maltoheptaose effectively. Subsequently, a mixture of raffinose and maltoheptaose (50:50% by weight) was filtered at 50°C. As shown in Figs. R6d and R6f, the mass content of raffinose in the permeate increased to 84%, indicating that the P-GOMs had a high selectivity towards raffinose and maltoheptaose at 50°C. Therefore, the P-GOMs have the performance of reliability and relatively high separation resolution.

In response to these comments, we have made revisions in the revised manuscript (page 20, line 352) and Supplementary Information (page 15, line 168).

Fig. R6 P-GOMs for raffinose, maltopentaose and maltoheptaose separation. a-c) The ion chromatogram of the original feed solution and the permeate obtained at 25°C and 50°C using raffinose, maltopentaose and maltoheptaose as the single solute. d) The ion chromatogram of the original feed solution and the permeate obtained at 50°C using a mixture of raffinose and maltoheptaose. e) The rejection rates of P-GOMs for raffinose, maltopentaose and maltoheptaose at 25°C and 50°C. f) Source and permeate content of a mixture of raffinose and maltoheptaose.

Comment 3:

Additionally, no rationale is given for the low concentrations that were used in the experiments. One is left to wonder whether those conditions are really relevant / how the material would perform if challenged with a more concentrated feed solution.

Response 3:

Thanks for this important comment. Following the reviewer's suggestion, we have performed the separation experiments using more concentrated feed solutions. We chose RB and CBB as examples to demonstrate the question. As shown in Fig. R7, the permeance decreased with the RB concentration increasing from 4 ppm to 200 ppm. The rejection rates for these four concentrations of RB were 99.6%, 93.5%, 93.1% and 91.1%, respectively. It can be seen that there was only a slight rejection rate decrease for RB with the increased concentration. For CBB, the anionic dye, the rejection rates were ~100%, 99.7% and 99.1% using 40 ppm, 100 ppm and 200 ppm feed solutions, and the permeance variation was small (Fig. R8). Therefore, the P-GOMs exhibited a good separation performance for a wide concentration range.

These results have been added in Supplementary Information (Supplementary Figure S21 in page 16, Supplementary Figure S21 in page 17 and the text in page 17, line 198).

Fig. R7 The separation performance of P-GOMs for RB solution with different concentration. a-d) The UV-vis absorption spectra before and after filtering RB from 25°C to 50°C with the concentrations of 4 ppm, 40ppm, 100ppm and 200 ppm. e-h) Temperature dependent permeances and rejection rates of P-GOMs for 4 ppm, 40ppm, 100ppm and 200 ppm RB.

Fig. R8 The separation performance of P-GOMs for CBB solution with different concentration. a) Temperature dependent permeances of P-GOMs for 40 ppm, 100ppm and 200 ppm CBB. b) Temperature dependent rejection rates of P-GOMs for 40 ppm, 100ppm and 200 ppm CBB.

Comment 4:

As an aside, it seems unlikely that highly confined water molecules in the nano-channels would behave like a bulk viscous flow (such that the Hagen-Poiseuille equation would be valid). It seems as though a more molecular approach to analyzing transport would be appropriate compared to a continuum approach.

Response 4:

We appreciate this important comment. The understanding of water flow behavior in the nano-channels of graphene oxide membrane is significant to explain the unusual high water permeance. Up to now, many researchers revealed that the water flow in the GO channel is viscous combined with experiments and classical molecular dynamics simulations [Nair, R., Wu, H., Jayaram, P., Grigorieva, I. & Geim, A. Unimpeded permeation of water through helium-leak-tight graphene-based membranes. *Science* **335**, 442-444 (2012); Huang, H. et al. Ultrafast viscous water flow through nanostrand-channelled graphene oxide membranes. *Nat. Commun.* **4**, 2979 (2013); Wei, N., Peng, X. & Xu, Z. Breakdown of fast water transport in graphene oxides. *Phys. Rev. E* **89**, 012113 (2014); Abozar, A. et al. Large-area graphene-based nanofiltration membranes by shear alignment of discotic nematic liquid crystals of graphene oxide. *Nat. Commun.* **7**, 10891 (2016)]. Therefore, we adopted this mechanism in our work. It is indeed that the behavior of confined water molecules in nano-channels plays an important role in fluid transportation. We would like to perform the molecular simulation in our follow-up work.

We have added the above references in the revised manuscript. (Refs. 29, 30, 47 and 48 in page 11, line 197)

Revision checklist of manuscript:

1. All the words “flux” were changed to “permeance”.
2. All the units “ $L m^{-2} h^{-1}$ ” were changed to “ $L m^{-2} h^{-1} bar^{-1}$ ”.
3. All the words “Absortance” were changed to “Absorbance”.
4. Page 11, line 197, the references “29, 30, 47, 48” were added.
5. Page 11, line 199, the reference “47” was deleted.
6. Page 14, line 241, the reference “36” was added.
7. Page 14, line 257, the sentence “For the P-GOMs at 25°C (relative humidity \approx 40%)” was changed to “For the P-GOMs in wet state”.
8. Page 14, line 259-260, the “ $2\theta = 5.43^\circ$ ” and “ $2\theta = 5.59^\circ$ ” were changed to “ $2\theta = 4.71^\circ$ ” and “ $2\theta = 5.91^\circ$ ”.
9. Page 15, line 262, the sentence “Besides, the entanglement of PNIPAM chains between two adjacent GO sheets also results in the decreased water permeance.” was added.
10. Page 15, line 264, the references “54, 55” were added.
11. Page 15, line 266, the “Fig. 4b” was changed to “Fig. 4c”.

12. Page 16, line 284, the “c)” was changed to “d)”.
13. Page 14, line 302, the “Fig. 4c” was changed to “Fig. 4d”.
14. Page 18, line 326, the references “22, 29, 57” were added.
15. Page 19, line 336, the “(Supplementary Fig. S19)” was added.
16. Page 20, line 352, the sentences “Besides, we also performed separation experiments using three non-ionic solutes: raffinose, maltopentaose and maltoheptaose (Supplementary Fig. S20). The results demonstrate that the P-GOMs the P-GOMs have the performance of reliability and high separation resolution.” were added.
17. Page 21, Fig. 5e, the “38.8%” was changed to “12.5%”.
18. Page 23, line 388-390, the sentences “and raffinose were all purchased from Aladdin Chemistry Co., Ltd. (Shanghai, China)” and “Maltopentaose and maltoheptaose (HPLC) were provided by Miragen (USA).” were added.
19. Page 26, line 454-455, the words “(Supplementary Fig. S21)” and “(Supplementary Fig. S22) were added.
20. Page 26, line 455, the sentence “200 mg/L raffinose, 200 mg/L maltopentaose and 200 mg/L maltoheptaose” was added.
21. Page 29, line 514, the sentence “The concentrations of raffinose, maltopentaose and maltoheptaose in the solution were analyzed by ion chromatography (IC) equipped with a CarboPacTM PA200 3X250 column.” was added.
22. Page 32, line 587, the reference 25 “Abraham, J. et al. Tunable sieving of ions using graphene oxide membranes. *Nat. Nanotechnol.* (2017).” was changed to “Abraham, J. et al. Tunable sieving of ions using graphene oxide membranes. *Nat. Nanotechnol.* **12**, 546-550 (2017).”

23. Page 37, line 683, the sentence “We are grateful for the financial support from the National Natural Science Foundation of China (21433012, 21374001, 21222309)” was changed to “We are grateful for the financial support from the National Natural Science Foundation of China (21774005, 21433012, 21374001, 21222309)”.
24. Page 40, line 747, the “c)” was changed to “d)”.

Revision checklist of Supplementary Information:

1. Page 5, line 51, the sentence “The G band at 1580 cm^{-1} was assigned to the vibration of sp^2 bonded carbon atoms, and the D band at 1350 cm^{-1} was assigned to the vibration of carbon atoms with dangling bonds in plane terminations of disordered graphite, indicating the formation of sp^3 carbon atoms.” was added.
2. Page 5, line 56, the “from 1580 cm^{-1} to about 1600 cm^{-1} ” was added.
3. Page 6, line 61, the sentence “P-GO show strong proton signals of PNIPAM units, indicating the polymerization of NIPAM on GO sheets.” was changed to “For PNIPAM polymer, the ^1H NMR spectra clearly shows the characteristic signals of PNIPAN side chains around $\delta = 1.0, 1.5, 1.9$ and 3.8 ppm. After the PNIPAM was grafted to GO sheets, diamagnetic ring currents in GO lead to the upfield shifts of the proton peaks in CH (b) and CH (d) groups, changing from $\delta = 3.8$ to $\delta = 3.6$ and $\delta = 1.9$ to $\delta = 1.8$. The appearance and upfield shifts of these PNIPAM characteristic signals demonstrates that the PNIPAM chains had been successfully grafted to the GO sheets.”
4. Page 8, Supplementary Figure S10c was added.
5. Page 8, line 85, the sentence “c) The survey scan XPS spectra of GO and GO-12 show the C/O atomic ratio increased from 2.06 (GO) to 2.36 (GO-12), demonstrating the reduction of GO.” was added.
6. Page 8, line 86, the “c)” was changed to “d)”.
7. Page 9, line 95, the sentences “Researchers had physically blended chemically reduced GO with PNIPAM to construct the thermal responsive graphene membrane, in which the chemically reduced GO and PNIPAM were connected by supramolecular interaction⁴. It showed a positive temperature-response gating behavior. For the P-GOM, it was composed of PNIPAM covalently bound GO via free radical polymerization. The covalently bound interactions between PNIPAM chains and GO endowed

the membrane negative temperature-response gating behavior.” were added.

8. Page 12, line 131, the word “flux” was changed to “permeance”.
9. Page 13-14, Supplementary Figure S19 and the related text were added.
10. Page 15-16, Supplementary Figure S20 and the related text were added.
11. Page 16-17, Supplementary Figure S21, Supplementary Figure S22 and the related text were added.
12. Page 21, line 271, the word “flux” was changed to “permeance”.
13. Page 21, in the Supplementary Table S2, the “N” was changed to “+”, “38.8” was changed to “12.5”, and the data of permeance were added.

REVIEWERS' COMMENTS:

Reviewer #1 (Remarks to the Author):

The authors have adequately addressed the concerns raised by the referees. I would like to recommend it for publication.

Reviewer #2 (Remarks to the Author):

The authors have well addressed my previous comments, I therefore do not have any further comments and support publication.

Reviewer #3 (Remarks to the Author):

I believe the points that I raised in the initial review have been satisfactorily addressed.

Response to Reviewers' comments:

Reviewer #1 (Remarks to the Author):

The authors have adequately addressed the concerns raised by the referees. I would like to recommend it for publication.

Reviewer #2 (Remarks to the Author):

The authors have well addressed my previous comments, I therefore do not have any further comments and support publication.

Reviewer #3 (Remarks to the Author):

I believe the points that I raised in the initial review have been satisfactorily addressed.

Response:

We thank all the reviewers for recommending our paper to be published without further revisions.